# Isolating the effects of carbohydrate and lipid availability on exercise-induced skeletal muscle signalling in males

Louise Bradshaw[1,2], Alfonso Moreno Cabañas[1,2,3], Bruno Spellanzon[1,2], Katie Marie Hutchins[1,2], Adam J. Collins[1,2], Jariya Buniam[1,2,4], Thibaux van der Stede[5], Joanne Mallinson[6], Kostas Tsintzas[6], Wim Derave[5], Jean-Philippe Walhin[1,2], James A. Betts[1,2], Francoise Koumanov[1,2] and Javier T. Gonzalez[1,2] (iD)

[1]*Centre for Nutrition, Exercise and Metabolism, University of Bath, Bath, UK*
[2]*Department for Health, University of Bath, Bath, UK*
[3]*Exercise Physiology Lab at Toledo, Faculty of Sport Sciences, University of Castilla–La Mancha, Toledo, Spain*
[4]*Princess Srisavangavadhana Faculty of Medicine, Chulabhorn Royal Academy, Bangkok, Thailand*
[5]*Department of Movement and Sport Sciences, Ghent University, Ghent, Belgium*
[6]*MRC Versus Arthritis Centre for Musculoskeletal Ageing Research, School of Life Sciences, University of Nottingham, Queen's Medical Centre, Nottingham, UK*

Handling Editors: Karyn Hamilton & Robert Musci

The peer review history is available in the Supporting Information section of this article (https://doi.org/10.1113/JP289864#support-information-section).

**Abstract figure legend** It is currently unknown whether the effects of low carbohydrate availability on exercise-induced skeletal muscle signalling and adaptations are due to the low carbohydrate availability per se, or the comcomittant high fatty acid availability. Comparing exercise after extended overnight fasting, with carbohydrate ingestion and with niacin ingestion allows for separation high carbohydrate availability from low fatty acid availability. This paradigm reveals that carbohydrate ingestion suppresses exercise-induced phosphorylation of human skeletal muscle acetyl-CoA carboxylase is independent of fatty acid availability, suggesting this signalling is more heavily influenced by carbohydrate availability per se.

The Journal of Physiology

**Abstract**  Training with low carbohydrate availability can increase AMP-activated protein kinase (AMPK) activation, but whether increased AMPK activation is the result of low carbohydrate availability *per se* or concurrent increases in fatty acid availability/oxidation is unclear. This study assessed the independent effects of carbohydrate and fatty acid availability on exercise-induced skeletal muscle AMPK activation and downstream signalling. Eight active males who were aged between 18 and 60 years with a body mass index in the range 18.0–30.0 kg m$^{-2}$ cycled on three occasions for 60 min at 95% of lactate threshold 1 with ingestion of either carbohydrate (CARB), niacin (NIACIN) or placebo (FAST) in a crossover design (11 ± 6 days washout). Blood and exhaled breath were sampled throughout exercise and muscle was sampled pre- and post-exercise. Fat oxidation and plasma non-esterified fatty acid concentrations were both lower in CARB *vs.* FAST with negligible difference between CARB *vs.* NIACIN. Plasma insulin concentrations were higher in CARB compared with both FAST and NIACIN. Net muscle glycogen use was greater with NIACIN *vs.* CARB. Although no evidence for differences were observed for phosphorylated AMPK, the downstream target, phosphorylated acetyl-CoA carboxylase was decreased with CARB *vs.* both FAST (−0.7 ± 0.6 fold, $P = 0.04$) and NIACIN (−1.0 ± 0.8 fold, $P = 0.02$). RNA-sequencing displayed several canonical changes with exercise but little difference between conditions. These data suggest carbohydrate ingestion suppresses exercise-induced phosphorylation of acetyl-CoA carboxylase independent of fatty acid availability.

(Received 4 August 2025; accepted after revision 23 October 2025; first published online 16 November 2025)

**Corresponding author** J. T. Gonzalez: Department for Health, University of Bath, Bath, BA2 7AY, UK.    Email: J.T.Gonzalez@bath.ac.uk

### Key points

- It is currently unknown whether the enhanced physiological adaptation to regularly exercising in a fasted-state are explained by low carbohydrate availability and/or the concomitant increase in fatty acid availability.
- This study used fasted exercise with niacin ingestion to reduce the lipaemic response associated with fasted exercise to isolate the effects of carbohydrate *vs.* fatty acid availability on exercise-induced skeletal muscle signalling.
- Our data show niacin ingestion increases muscle glycogen utilisation compared to carbohydrate ingestion during exercise, but both niacin and carbohydrate ingestion suppress fatty acid availability and fat oxidation to a similar extent.
- Our data demonstrate carbohydrate ingestion during exercise suppresses acetyl-CoA carboxylase phosphorylation compared to both niacin ingestion and extended overnight fasting.
- These data suggest that high carbohydrate availability inhibits exercise-induced acetyl-CoA carboxylase phosphorylation in human skeletal muscle, independent of circulating fatty acid concentrations.

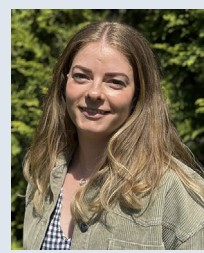

**Louise Bradshaw** is a postdoctoral researcher in the Centre for Nutrition, Exercise and Metabolism at the University of Bath. Her research focuses on exercise timing and temporal regulation of metabolism in humans. Her PhD work focused on the effects of substrate use during exercise on skeletal muscle metabolism and risk of cardiometabolic diseases. Prior to her PhD she completed a medical degree and master's degree in Human Nutrition specialising in Sport and Exercise Nutrition at the University of Glasgow and worked in clinical medicine for several years. She also has a specialist interest in female physiology and exploring sex differences.

## Introduction

Skeletal muscle adaptations to regular exercise are considered to be largely related to the accumulation of signalling events from each exercise bout. A major mechanism regulating such adaptations is the key energy sensing pathway involving AMP-activated protein kinase (AMPK). During exercise, the increase in AMP/ATP ratio activates AMPK through a series of binding and phosphorylation events (Mounier et al., 2015). Phosphorylated AMPK (pAMPK) activates several other pathways; for example, those responsible for increased glucose uptake via GLUT4 translocation; increased fatty acid uptake and oxidation via inhibition of acetyl-CoA carboxylase (ACC); and enhanced contractile activity and mitochondrial biogenesis (Spaulding & Yan, 2022).

As a result o fthe key role of AMPK in adaptations to endurance exercise training, research has focused on nutritional interventions to enhance AMPK activation in response to each acute bout of exercise. Recent research has focused on restricting carbohydrate availability before, during and/or after exercise as a strategy to augment exercise-induced skeletal muscle AMPK activation and metabolic adaptations (Edinburgh et al., 2020, 2021; Impey et al., 2018). Data suggest that training with lower glycogen concentrations may increase transcriptional accumulation of genes encoding key mitochondrial proteins such as *PDK4*, *CPT1* and *PCG1A*, at the same time as increasing cell signalling and oxidative enzyme activity (Hansen et al., 2005; Pilegaard et al., 2005; Steinberg et al., 2006; Yeo et al., 2008).

To date, the strategies used to manipulate carbohydrate/glycogen availability have involved fasting, carbohydrate restriction and/or exercise resulting in simultaneously increasing the availability of lipid substrates, such as non-esterified fatty acid (NEFA) (Akerstrom et al., 2006; De Bock et al., 2008; Hulston et al., 2010; Nybo et al., 2009; Pilegaard et al., 2005; van Proeyen et al., 2011). It is important to isolate the effect of muscle glycogen from NEFA concentrations because NEFA concentrations may play an independent role in muscle cell signalling and adaptation. In rodents, a combination of daily heparin injections and a high fat diet resulted in an increase in plasma NEFA availability, skeletal muscle mitochondrial DNA copy number and enzymes of the fatty acid pathway, citric cycle and respiratory chain, suggesting an increase in mitochondrial biogenesis (Garcia-Roves et al., 2007). Further work has demonstrated that infusion of palmitate for 6 days in rodents leads to an increase in pAMPK and phosphorylated ACC (pACC) potentially via the calcium-dependant pathway and calcium/calmodulin-dependant protein kinase (Anavi et al., 2010). Therefore, to date, no study has teased apart the independent effect of muscle glycogen concentrations from circulating NEFA.

The present study aimed to assess the effects of carbohydrate independent of fatty acid availability on exercise-induced skeletal muscle AMPK activation. To isolate carbohydrate from fatty acid availability, we compared three conditions: exercise in the overnight fasted-state (intended to result in relatively high fatty acid availability with relatively low carbohydrate, insulin and glycogen availability); carbohydrate ingestion (intended to result in low fatty acid availability and relatively high carbohydrate availability); and fasted exercise with niacin ingestion, which acts on receptor GPR109A to reduce fatty acid mobilisation in adipose tissue (Christie et al., 1996) to reduce fatty acid availability with relatively low carbohydrate and insulin availability. We hypothesised that reduced carbohydrate availability will be more important than high fatty acid availability for AMPK signalling and therefore fasted-state exercise and niacin will result in greater AMPK signalling compared to carbohydrate ingestion.

## Methods

### Ethical approval

All participants provided written informed consent and the study was conducted in accordance with the latest version of the *Declaration of Helsinki*. Ethical approval was provided by the National Health Service Research Ethics Committee Bristol (22/SW/662). The current data are part of a larger investigation examining the impact of lipolysis on energy intake following exercise which is registered at clinicaltrials.gov (NCT05417659).

### Participants and study design

The study was a randomised, cross-over experiment. Participants attended the laboratory on four occasions for preliminary testing and three laboratory visits. The conditions for the main laboratory visits were fasted-state exercise (FAST), exercise with carbohydrate (CARB) and exercise with niacin (NIACIN). Participant characteristics are shown in Table 1. Participants were healthy males aged between 18 and 60 years with a body mass index in the range 18.0–30.0 kg m$^{-1}$ and recreationally active (at least 30 min of exercise three times a week). Prospective participants were excluded if they had any medication or medical condition that could introduce bias to the study or pose personal risk to them, any sleeping disorder or sleep apnoea, a contradiction to niacin, or weight instability defined as greater than 5 kg change in body mass over the last 6 months. During screening eligible participants were

**Table 1. Participant characteristics**

| Characteristic | Value |
| --- | --- |
| Age (years) | $30 \pm 13$ |
| Height (m) | $1.8 \pm 0.1$ |
| Body mass (kg) | $75.4 \pm 6.2$ |
| Body mass index (kg m$^{-2}$) | $23.1 \pm 2.9$ |
| Fat mass (kg) | $13.0 \pm 4.3$ |
| Fat free mass (kg) | $62.2 \pm 2.7$ |
| Self-reported physical activity rating | $9 \pm 3$ |
| Lactate threshold 1 power (W) | $114 \pm 24$ |

Data are shown as the mean $\pm$ SD ($n = 8$). Physical activity rating scale taken from Jamnick et al. (2018).

given a dose of niacin (5 mg kg$^{-1}$) to ensure no adverse reaction and to familiarise participants with potential side effects such as flushing. Following screening, participants attended the laboratory for preliminary testing prior to three main laboratory visits.

### Preliminary testing

Participants arrived at the laboratory in an overnight fasted state having abstained from exercise and alcohol for 24 h prior. Participants recorded all dietary intake during the 24 h prior to preliminary testing and were asked to repeat this intake prior to all subsequent visits to ensure participants were in a similar state of energy balance for all trials. Height was measured using a stadiometer (Seca Ltd, Hamburg, Germany) in the Frankfurt plane with participants barefoot. Weight was measured using digital scales (Tanita, Amsterdam, The Netherlands). A dual-energy X-ray absorptiometry scan (Hologic, Marlborough, MA, USA) was performed with participants lying supine wearing light clothing to determine fat mass (kg) and fat-free mass (kg).

Participants were fitted with a heart rate monitor (Polar Electro Oy, Kempele, Finland) and performed a submaximal exercise test to determine lactate threshold 1 (LT1) on an electronically-braked cycle ergometer (Lode Corival, Groningen, The Netherlands). The exercise test consisted of nine 4 min incremental stages with the intensity of each stage individualised to each participant according to the method of Jamnick et al. (2018). Baseline heart rate and blood lactate concentrations were recorded prior to starting the exercise test. During the last minute of each stage a 1 min expired breath sample was collected using the Douglas bag technique and capillary blood sample was obtained for determination of blood lactate concentration using a handheld monitor (Lactate Plus; Nova Biomedical, Runcorn, UK). Heart rate and rating of perceived exertion [RPE; Borg (1970)] were recorded at the end of each stage. LT1 was determined by an increase

in lactate concentration of 0.5 mmol L$^{-1}$ above resting levels.

### Main laboratory visits

Participants attended the laboratory after a $11 \pm 6$ day washout period after an overnight fast and dietary standardisation. A schematic of main laboratory visits is shown in Fig. 1. On arrival at the laboratory, weight was measured and a cannula was inserted into an antecubital fossa vein on the left arm. A baseline blood sample was obtained and the cannula flushed with 10 mL of saline (B Braun, Melsungen, Germany). A muscle biopsy sample was taken from the vastus lateralis using a Bergstrom needle attached to suction under local anaesthetic (lidocaine 1%; Hamelm, Gloucester, UK). A further incision was made ~3 cm from to the first biopsy site in preparation for the next biopsy. The muscle tissue was snap frozen in liquid nitrogen before being stored at –70°C until analysis. The side which biopsies were obtained during the first visit was randomised by dominant or non-dominant leg and biopsies were taken from alternating sides for each subsequent visit.

Following the muscle biopsy, participants consumed either a placebo drink (FAST and NIACIN visits) or carbohydrate drink containing 1.6 g kg$^{-1}$ body mass of maltodextrin (CARB visit; MyProtein, Manchester, UK) in 400 mL. The placebo drink was matched for taste and volume to the carbohydrate drink and contained sucralose and flavouring (MyProtein). Participants rested supine for 1 h prior to commencing exercise. After 30 min participants consumed either placebo tablets (CARB and FAST visits; lactose, sucrose and magnesium stearate; Homeopathic Supply Company, Bodham, UK) or niacin tablets (NIACIN visit; 10 mg kg$^{-1}$; Solgar, Leonia, NJ, USA). After 60 min of rest a digital heart rate monitor was fitted and participants cycled at 95% LT1 for 60 min. The intensity of exercise was chosen as LT1 correlates with maximal fat oxidation (Achten & Jeukendrup, 2004), allowing the greatest difference in substrate oxidation between FAST and CARB conditions. Participants consumed further drinks (100 mL) at the onset of exercise and every 15 min with each dose delivering 0.2 mg kg$^{-1}$ maltodextrin or placebo. At the onset of exercise and after 30 min participants consumed further doses of niacin (0.5 mg kg$^{-1}$ per dose) or placebo. Blood samples and a 1 min expired breath sample were collected at 15, 30, 45 and 60 min of cycling. Indirect calorimetry was used to analyse expired breath samples and the rate of fat and carbohydrate oxidation were calculated using stoichiometric equations (Frayn, 1983). Within 5 min of completion of exercise a further muscle biopsy was taken from the vastus lateralis using a Bergstrom needle attached to suction.

## Plasma metabolites and hormones

Blood samples were centrifuged for 10 min at 4000 **g** and 4°C and the plasma collected. Samples were stored at −70°C until analysis. Samples were analysed for plasma glucose, lactate and NEFA concentrations using an automated analyser (Daytona RX; Randox Laboratories Ltd, Crumlin, UK). Plasma insulin concentrations were analysed using a commercially available enzyme-linked immunosorbent assay according to the manufacturer's instructions (Mercodia, Uppsala, Sweden). Inter-assay CV were <3% for glucose, <1% for glucose, <2% for NEFA, and intra- and inter-assay CV for insulin were <7% and <9%, respectively.

## Muscle glycogen concentrations

Samples were freeze-dried overnight under vacuum at −55°C (Mechatech Systems, Bristol, UK). Following removal of visible connective tissue and blood contamination, samples were ground to a powder using a pestle and mortar and stored at −20°C until analysis. Then, ∼2–5 mg of powder was digested in 0.1 mM NaOH and neutralised with HCl-citrate buffer solution (at pH 5.0). Glycogen was then hydrolysed with $\alpha$-amyloglucosidase before triplicate analysis of glycosyl units on a spectrophotometric plate reader (SpectraMax 190; Molecular Devices, San Jose, CA, USA) using a method previously described (Harris et al., 1974). Glycogen concentrations are reported as millimoles glucosyl units per kilogram of dry muscle mass (mmol kg$^{-1}$ DM).

## Muscle western blotting

Powdered muscle samples were solubilised in 100 μL mg$^{-1}$ dry powder RIPA buffer (50 mM Tris, pH 7.4, 150 mM NaCl, 0.5% sodium deoxycholate; 0.1% SDS and 0.1% NP-40), supplemented with protease and phosphatase inhibitors (Protease and Phosphatase Inhibitor Cocktail; Thermo Fisher Scientific, Waltham, MA, USA) and the protein concentration of each sample was determined using a bicinchoninic acid protein assay kit (Thermo Fisher Scientific). An equal amount of protein (50 μg) was loaded into each well in 10% Tris-glycine SDS-polyacrylamide gels and proteins were separated using SDS-PAGE at 200 V. Proteins were transferred to a nitrocellulose membrane via semi-dry electro-blotting apparatus (Bio-Rad, Hercules, CA, USA). The membranes were washed in Tris-buffered saline (0.09% NaCl, 100 mM Tris-HCl pH 7.4) with 0.1% Tween 20 (TBS-T) and incubated for 30 min in a blocking solution of 5% non-fat dried milk (Marvel; Premier Foods, Dublin, Ireland) in TBS-T. Membranes were washed again in TBS-T and incubated in primary antibodies against phospho Ser 79 acetyl-CoA carboxlyase (ACC) (Cell Signaling Technologies, Danvers, MA, USA), phospho Thr172 AMP-dependent protein kinase (AMPK) (Cell Signaling Technologies), phospho-Thr642 Akt substrate of 160 kDa (AS160) (Cell Signaling Technologies), phospho-Tyr1150/1151 insulin receptor (INS) (Cell Signaling Technologies), phospho-Ser 473 protein kinase B (Akt) (Cell Signaling Technologie), phospho-Ser660 hormone-sensitive lipase (HSL) (Cell Signaling Technologies), phospho-Ser 293 pyruvate dehydrogenase (PDH) (Cell Signaling Technologies), vinculin (Cell Signaling Technologies), pyruvate dehydrogenase kinase 4 (PDK4) (Abgent, San Diego, CA, USA) and beta-actin (Protein Technologies, Stockport, UK) overnight at 4°C. In the morning, membranes were washed in TBS-T and incubated in a secondary antibody (1:4000 dilution of anti-species IgG horseradish peroxidase-conjugated antibody in the blocking solution) for 1 h. The nitrocellulose membranes were again washed with TBS-T and incubated

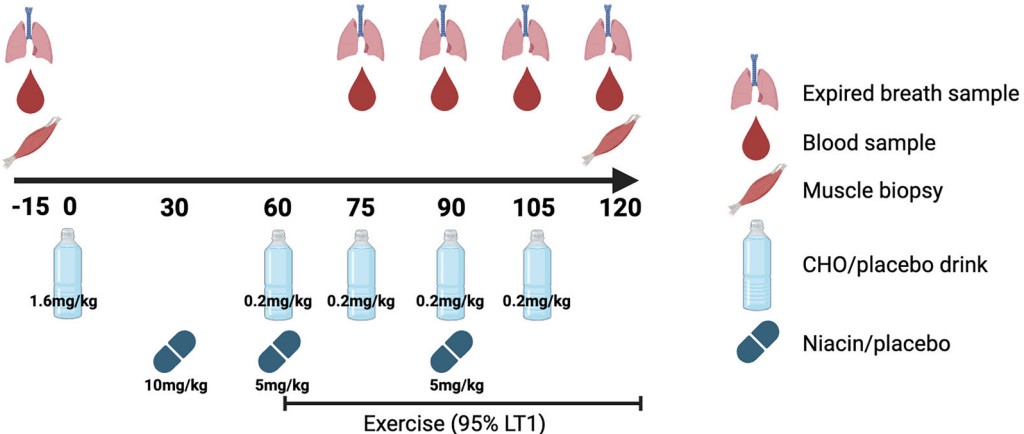

**Figure 1. Laboratory visits**
Schematic of each of the three laboratory visits.

in an enhanced chemiluminescence reagent before being visualised using a chemiluminescent imager (EpiChem II Darkroom; UVP, Upland, CA, USA). The nitrocellulose membranes were incubated in stripping solution (62.5 mM Tris, pH 6.7, 2% SDS) with 350 µL of $\beta$-mercaptoethanol at 52°C before reblotting [ACC, AMPK, AS160, INSR, Akt, HSL and PDH (Cell Signaling Technologies), GLUT4 (in-house C-terminal GLUT4 antibody provided by Dr Françoise Koumanov) (Satohs et al., 1993)]. All samples from each participant were run on the same gel. Band densities were quantified using Image Studio Lite, version 5.2.5 (LI-COR Biosciences, Cambridge, UK). The fold-change in phosphorylation status (ratio phosphorylated to total) or total protein content from pre- to post- exercise was calculated.

### Gene expression

Samples were homogenised in 1 mL of QIAzol lysis reagent (Qiagen, Crawley, UK) and 200 µL of chloroform was added. The samples were centrifuged and the upper aqueous phase containing the total RNA was collected mixed with 1.5 volume of ethanol. The total RNA was purified using a miRNeasy RNA extraction kit (Qiagen) in accordance with the manufacturer's instructions. mRNA sequencing was performed by Novogene (Amsterdam, The Netherlands). mRNA was purified from total RNA using poly-T oligo-attached magnetic beads. After fragmentation, the first strand cDNA was synthesised using random hexamer primers, followed by the second strand cDNA synthesis using either dUTP for directional library or dTTP for non-directional library. The library was checked with Qubit (Thermo Fisher Scentific) and real-time PCR for quantification and Bioanalyser (Agilent Technologies, Santa Clara, CA, USA) for size distribution detection. Sequencing was performed in paired-end mode with a read length of 150 bp using Illumina X Plus PE150 technology (Illumina, Inc., San Diego, CA, USA). Clean reads were obtained by removing reads containing adapter, reads containing poly-N and low quality reads from raw data. Paired-end reads were aligned on the reference genome using Hisat2, version 2.0.5. The mapped reads of each sample were assembled using StringTie, version 1.3.3b.

Mapped count data was then read into R, version 4.4.2 (R Foundation, Vienna, Austria) for all further analyses. Only participants with at least a pre- and post-exercise sample for two out of three experimental conditions were kept in the dataset ($n = 7$ participants). The data was then transformed into a DESeqDataSet object via DESeq2, version 1.46.0. Only genes with at least 10 counts in at least seven samples were retained for downstream analysis ($n = 4714$ genes). No apparent outlier samples were identified via distance plots on variance stabilised data or principal component analysis plots on the top 500 variable genes. Random technical variation was included in the statistical model via modelling surrogate variables ($n = 2$) using the sva package (version 3.54.0). Differential gene expression analysis was then performed using the DESeq() and results() functions in the DESeq2 pipeline. DE genes (adjusted $P < 0.05$ and absolute $\log_2$-foldchange > 0.25) across the three conditions were compared using the eulerr package (version 7.0.2).

### Statistical analysis

Our previous data showed exercise in a fed-state resulted in an increase in pACC (ratio pACC to total ACC) of 7.47 $\pm$ 6.57 *vs.* 3.81 $\pm$ 2.49 after exercise in a fasted-state (Edinburgh et al., 2018). Using the effect size of $d = 1.4$, testing eight participants would provide a beta of 0.9 with an $\alpha$-level of 0.05. Prism, version 10.2.3 (GraphPad Software Inc., San Diego, CA, USA) was used for statistical analysis and data are presented as the mean $\pm$ SD. Data were checked for normality by visual inspection of Q-Q plots. To retain statistical power in this exploratory analysis, comparisons were limited to differences between two conditions in line with the principle of closed testing (Bender & Lange, 2001): FAST *vs.* CARB and CARB *vs.* NIACIN. Differences between conditions in substrate oxidation, glycogen utilisation and protein content were analysed by a paired $t$ test. For plasma metabolite concentrations, heart rate and RPE interactions were analysed by two-way ANOVA to assess differences in time and condition. A one-way ANOVA was used to assess differences in substrate oxidation, mean $\dot{V}_{O_2}$ and energy expenditure. Multiple comparisons were assessed by Tukey's test. The difference between CARB *vs.* FAST and CARB *vs.* NIACIN in glycogen utilisation (changes from pre- to post-exercise concentrations) and pACC was calculated and the association was assessed by Pearson's product-moment correlation coefficients. Size of correlations were defined as small for an $r$ value of 0.10 to 0.30, moderate for 0.31 to 0.60 and large >0.6. $P < 0.05$ was considered statistically significant.

### Results

#### Metabolic responses

Plasma NEFA concentrations decreased at onset of exercise in all three conditions but more so in CARB and NIACIN. Plasma NEFA concentrations then increased during exercise in FAST and remained suppressed in CARB and NIACIN with significantly higher concentrations in FAST at 75 min (FAST *vs.* CARB, $P = 0.008$; FAST *vs.* NIACIN, $P = 0.03$), 90 min (FAST *vs.* CARB, $P = 0.02$; FAST *vs.* NIACIN, $P = 0.03$) and 105 min (FAST *vs.* CARB, $P = 0.03$; FAST *vs.* NIACIN, $P = 0.04$). There was a significant effect of time ($P < 0.001$) and

condition ($P < 0.001$) on plasma NEFA concentrations and a time by condition interaction ($P < 0.001$) (Fig. 2*A*).

Plasma insulin concentrations increased in CARB at the onset of exercise and remained elevated throughout exercise with higher concentrations compared to FAST and NIACIN at 75 min (FAST, $P = 0.01$; NIACIN, $P = 0.02$), 90 min (FAST, $P = 0.03$; NIACIN, $P = 0.04$) and 105 min (FAST, $P = 0.02$; NIACIN, $P = 0.04$). Plasma insulin concentrations reduced at the onset of exercise in FAST and continued to decline during exercise. In NIACIN, plasma insulin concentrations remained similar throughout the protocol. There was a significant difference in plasma insulin concentrations between conditions (time × condition, $P < 0.001$) (Fig. 2*B*).

At onset of exercise plasma glucose concentrations fell in CARB and were significantly different to FAST and NIACIN (FAST, $P = 0.03$; NIACIN, $P = 0.007$). Plasma glucose concentrations then increased during exercise in CARB with no detectable differences between conditions. There was a significant difference in plasma glucose concentrations between conditions (time × condition, $P < 0.001$) (Fig. 2*C*). Plasma lactate concentrations increased at onset of exercise in all conditions (effect of time, $P < 0.001$) (Fig. 2*D*) but there were no significant differences between conditions (time × condition, $P = 0.38$). These metabolic responses confirm the conditions were successful in modulating NEFA and carbohydrate availability.

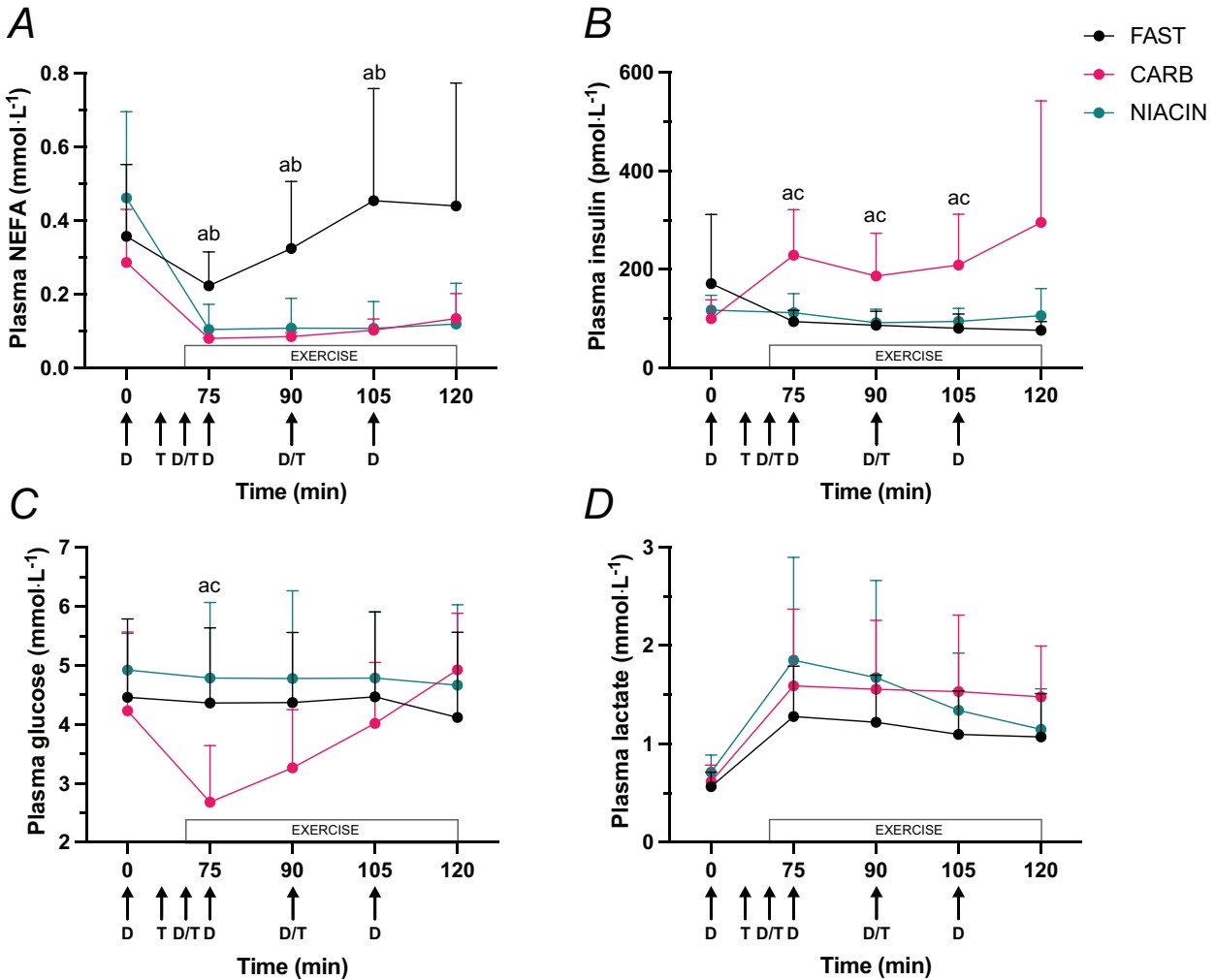

**Figure 2. Plasma non-esterified fatty acid (NEFA), insulin, glucose and lactate concentrations**
Plasma NEFA (*A*), insulin (*B*), glucose (*C*) and lactate (*D*) concentrations at baseline and during 1 h of exercise when performed in either an overnight fasted state (FAST) or with ingestion of carbohydrate (CARB) or niacin (NIACIN) before and during exercise. Data are presented as the mean ± SD. Differences between conditions assessed using two-way ANOVA and multiple comparisons assessed with Tukey's test. White box indicates exercise. Arrow with a 'D' indicates drink ingestion. Arrow with a 'T' indicates tablet ingestion. [a]FAST *vs.* CARB, $P < 0.05$, [b]FAST *vs.* NIACIN, $P < 0.05$, [c]CARB *vs.* NIACIN, $P < 0.05$ ($n = 8$ males).

**Table 2. Mean heart rate, RPE and $\dot{V}_{O_2}$, and total energy expenditure over 1 h of cycling at 95% LT1.**

|  | FAST | CARB | NIACIN |
|---|---|---|---|
| Heart rate | 123 ± 15 | 133 ± 11 | 132 ± 19 |
| RPE | 11 ± 1 | 12 ± 1 | 12 ± 1 |
| $\dot{V}_{O_2}$ (mL kg$^{-1}$ min$^{-1}$) | 24.18 ± 3.61 | 23.81 ± 3.33 | 23.29 ± 3.34 |
| Total energy expenditure (kcal) | 517 ± 99 | 500 ± 89 | 502 ± 66 |

Data shown as the mean ± SD (*n* = 8).

## Substrate oxidation

Total fat oxidation during exercise was increased in FAST compared to CARB (mean ± SD; FAST 22 ± 9 g *vs.* CARB 13 ± 8 g, $P = 0.003$) (Fig. 3*A*) but there were no detectable differences between CARB and NIACIN (NIACIN 15 ± 8 g, $P = 0.59$) (Fig. 3*A*). CARB significantly increased total carbohydrate oxidation compared to FAST (FAST 80 ± 21 g *vs.* CARB 96 ± 24 g, $P = 0.045$) (Fig. 3*B*) with no further increase with NIACIN (NIACIN 93 ± 27 g, $P = 0.76$ *vs.* CARB) (Fig. 3*B*). CARB significantly increased mean RER during exercise compared to FAST (FAST 0.88 ± 0.04 *vs.* CARB 0.93 ± 0.04, $P < 0.001$) (Fig. 3*C*) with no further increase compared to NIACIN (NIACIN 0.92 ± 0.06, $P = 0.7$) (Fig. 3*C*).

## Heart rate and RPE

Mean heart rate, RPE and $\dot{V}_{O_2}$, and total energy expenditure over the 1 h of exercise are shown in Table 2. Heart rate increased during exercise (effect of time $P < 0.001$) but there was no detectable difference between conditions (effect of condition $P = 0.35$; time × condition $P = 0.07$). Similarly, RPE also increased during exercise (effect of time $P < 0.001$) but there was no detectable difference between conditions (time × condition $P = 0.19$). There were no significant differences between energy expenditure ($P = 0.91$) or $\dot{V}_{O_2}$ ($P = 0.88$) between conditions.

## Muscle glycogen utilisation

There was a significant decrease in muscle glycogen concentration pre- to post-exercise in each condition (mean difference ± SD; FAST –97 ± 97 g, $P = 0.03$; CARB –58 ± 56 g, $P = 0.02$; NIACIN –142 ± 76 g, $P = 0.003$) (Fig. 4*A*). Net muscle glycogen utilisation (change from pre- to post-exercise) was not detectably different between FAST and CARB (mean ± SD; FAST –97 ± 97 mmol kg$^{-1}$ DM, CARB –58 ± 56 mmol kg$^{-1}$ DM, $P = 0.27$) (Fig. 4*B*), whereas NIACIN increased net glycogen utilisation relative to CARB (NIACIN –142 ± 76 mmol kg$^{-1}$ DM, $P = 0.01$) (Fig. 4*B*).

## Muscle protein content and signalling pathways activation

We examined total content and phosphorylation status of proteins post- to pre-intervention in the insulin

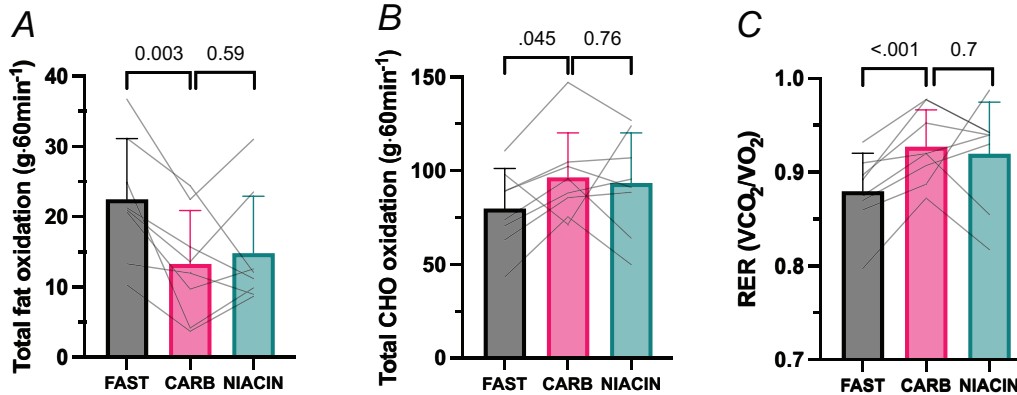

**Figure 3. Total fat and carbohydrate oxidation and mean respiratory exchange ratio (RER)**
Total fat (*A*) and carbohydrate (*B*) oxidation and mean RER) (*C*) during 1 h of exercise performed in either an overnight fasted state (FAST) or with ingestion of carbohydrate (CARB) or niacin (NIACIN) before and during exercise. Data are presented as the mean ± SD. Differences between FAST *vs.* CARB and CARB *vs.* NIACIN assessed using a paired *t* test (*n* = 8 males).

and AMPK signalling, and glycolysis pathways (Fig. 5). Compared to FAST there was no significant difference of pINSR/INSR (mean difference ± SD; −-0.3 ± 0.1 fold, $P = 0.33$) or pAS160/AS160 (−0.07 ± 0.6 fold, $P = 0.83$) in the CARB group. There was also no significant difference of pINSR/INSR or pAS160/AS160 between CARB and NIACIN (INSR 0.1 ± 0.2 fold, $P = 0.9$; AS160 0.007 ± 0.5 fold, $P = 0.88$). Phosphorylation status of Akt was significantly increased in CARB compared to FAST (0.8 ± 0.4 fold, $P < 0.003$) but not NIACIN (−0.6 ± 0.6 fold, $P = 0.08$). Total GLUT4 protein content pre- to post-exercise was not meaningfully different in

CARB compared to FAST (0.1 ± 0.5 fold; $P = 0.62$) or NIACIN (−0.08 ± 0.4 fold, $P = 0.54$).

There was no significant difference in pAMPK between CARB and FAST (−0.2 ± 0.5 fold, $P = 0.65$) or NIACIN (0.2 ± 0.6 fold, $P = 0.37$), although, as expected with an exercise intervention in all the conditions, the pAMPK Thr172/AMPK ratios were higher post intervention. The AMPK downstream effector pACC/ACC ratios were also elevated; however, in CARB, a significantly lower ratio pACC/ACC was observed compared to FAST (−0.7 ± 0.6 fold, $P = 0.04$) and NIACIN (-1.0 ± 0.8 fold, $P = 0.02$).

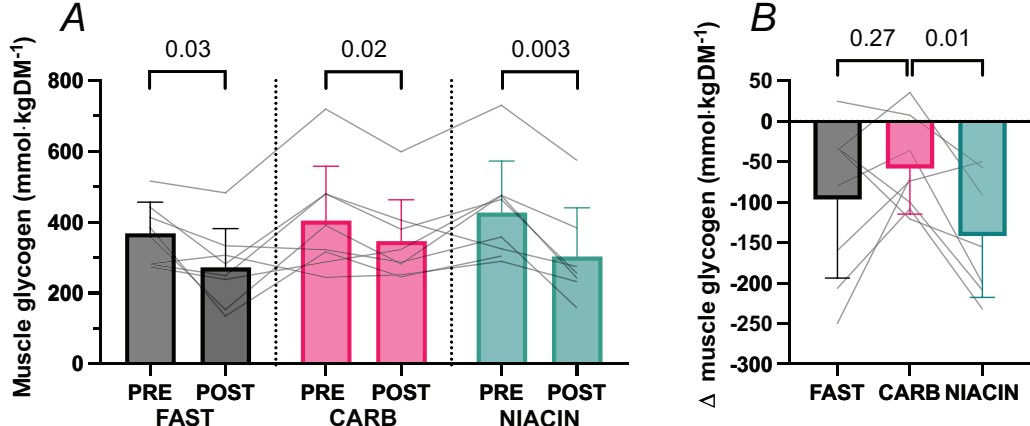

**Figure 4. Muscle glycogen concentrations and change in glycogen concentrations pre- and post-exercise**
Muscle glycogen concentrations pre- and post-exercise (*A*) and change in glycogen concentrations from pre- to post-exercise (*B*). Data shown as the mean ± SD. Differences between FAST *vs.* CARB and CARB *vs.* NIACIN assessed using a paired *t* test (*n* = 8 males).

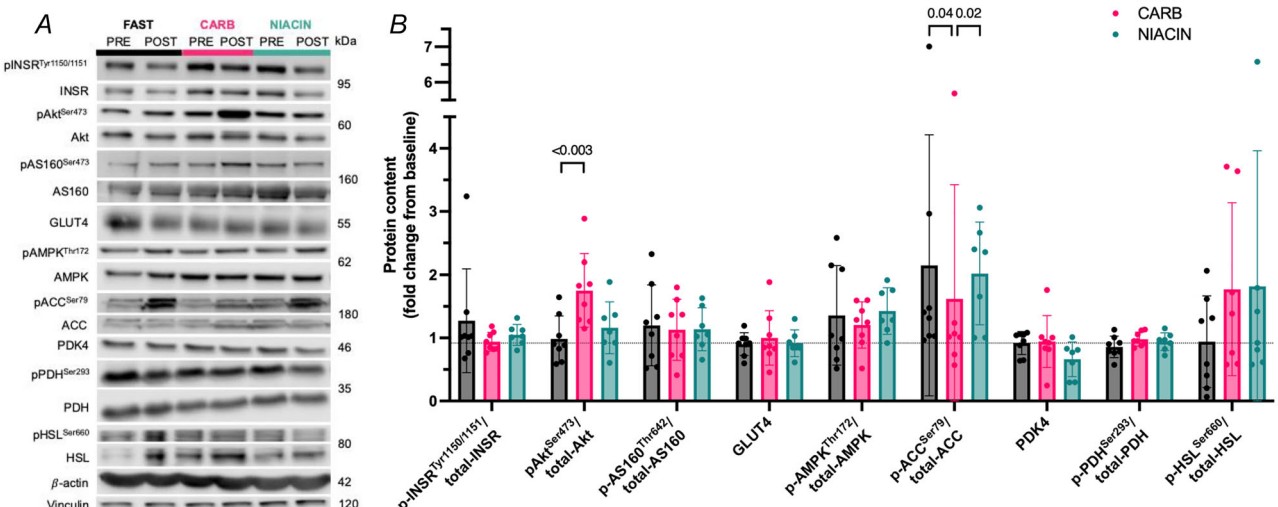

**Figure 5. Fold change of phosphorylation status and total protein content pre- to post-exercise**
Fold change of phosphorylation status and total protein content pre- to post-exercise of INSR$^{Tyr1150/1151}$, Akt$^{Ser473}$, AS160$^{Thr642}$, GLUT4, AMPK$^{Thr172}$, ACC$^{Ser79}$, PDK4, PDH$^{Ser293}$ and HSL$^{Ser660}$ (*B*) with representative blots (*A*). Data shown as the mean ± SD. Differences between FAST *vs.* CARB and CARB *vs.* NIACIN assessed using a paired *t* test (*n* = 8 males).

CARB resulted in no significant difference in pPDH/PDH compared to FAST ($0.1 \pm 0.2$ fold, $P = 0.14$) or NIACIN ($-0.04 \pm 0.1$ fold, $P = 0.37$). Similarly, there were no meaningful differences of PDK4 content of CARB compared to FAST ($0.02 \pm 0.3$ fold, $P = 0.88$) or NIACIN ($-0.3 \pm 0.4$ fold, $P = 0.19$). CARB resulted in no significant differences in pHSL/HSL compared to FAST ($0.9 \pm 1.6$ fold, $P = 0.2$) or NIACIN ($0.05 \pm 2.1$ fold, $P = 0.6$).

### Association between muscle glycogen utilisation and ACC phosphorylation

The difference between CARB and FAST of muscle glycogen utilisation and pACC displayed a moderate negative correlation ($r = -0.42$, $P = 0.35$) but not significant. There was only a small, positive correlation between CARB and NIACIN conditions which was also not significant ($r = 0.17$, $P = 0.75$).

### Gene expression

Illumina bulk RNA sequencing was performed to assess potential changes in the transcriptional response following exercise across the three conditions. An average of 9.8 million counts were detected per sample in a homogenous distribution across samples, with a minimal variability in identified genes across samples (Fig. 6*A*). Exercise was associated with a clear, but heterogenous, transcriptional perturbation in the muscle biopsies based on the principal component analysis (Fig. 6*B*). Differential gene expression (DEG) analysis identified 96–198 DEGs per condition, with overall less DEGs in FAST compared to CARB and NIACIN (Fig. 6*C*). This suggests that there are both shared DEGs across conditions and apparent condition-specific DEGs. Shared DEGs included canonical exercise- and stress-responsive genes (e.g. *NR4A3*, *EGR1*, *MAFF* and *JUNB*). Genes upregulated in CARB and NIACIN, but not FAST, included *PER2*, *HIF3A* and *XIRP1*. When comparing the magnitude of exercise-induced changes between the three conditions, only very subtle differences were observed (Fig. 6*D*).

### Discussion

To date, no study has directly assessed the effects of fatty acid availability independent of carbohydrate and insulin availability on exercise-induced AMPK signalling in human skeletal muscle. With the use of niacin, we were able to reduce fatty acid availability during exercise without increasing exogenous carbohydrate

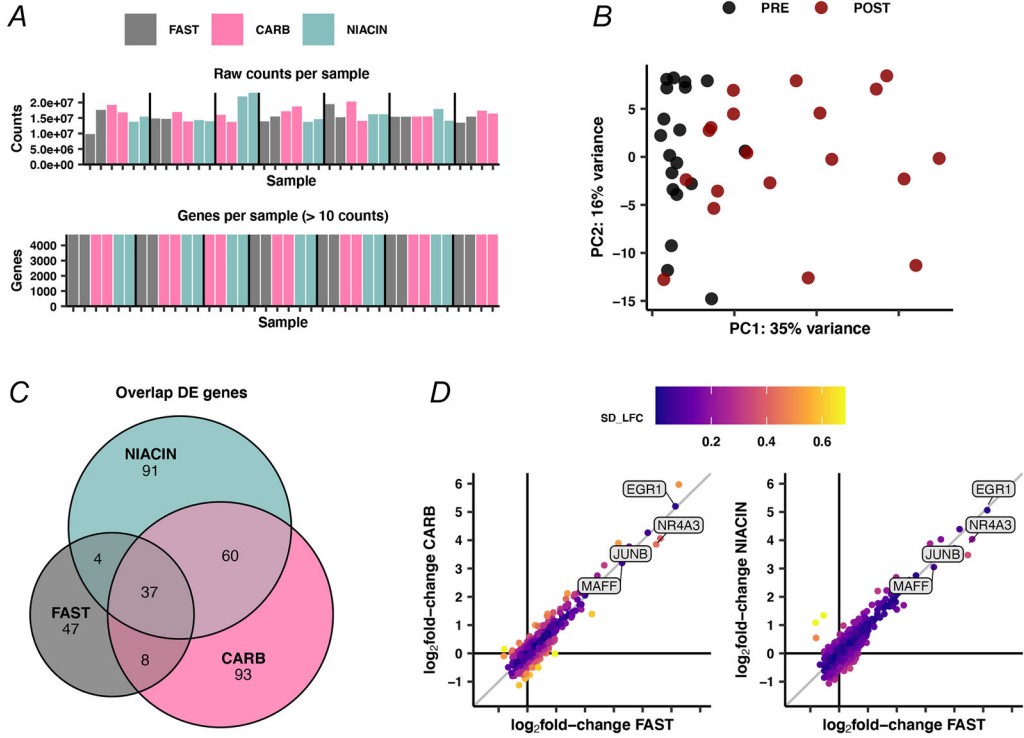

**Figure 6. RNA-sequencing (RNA-seq) quality control plots**
RNA-seq quality control plots showing number of counts and identified genes for each sample (*A*), principal component analysis (*B*), Venn diagram of differentially expressed genes (*C*) and log$_2$fold-change from pre- to post-exercise comparisons across conditions (*D*) (*n* = 7 males).

(energy) availability or insulin concentrations. These data demonstrate that carbohydrate (energy) ingestion reduces net muscle glycogen utilisation and exercise-induced ACC phosphorylation compared to niacin ingestion, despite similar circulating NEFA concentrations. This suggests that exercise-induced ACC inactivation may be sensitive to carbohydrate availability and muscle glycogen utilisation, independent of fatty acid availability.

Differences in substrate metabolism and availability have been implicated in exercise-signalling in skeletal muscle. One of the most researched signalling pathways in response to exercise is via activation of pAMPK, which subsequently phosphorylates and inactivates ACC to favour catabolic pathways activation and energy store replenishment. *In vitro*, increased fatty acid availability in the form of the long chain fatty acid, palmitate, increases AMPK and ACC phosphorylation (Fediuc et al., 2006). However, AMPK activation may not be essential for increased ACC posphorylation. AMPK $\beta 2$ knockout rodents still demonstrate increased fat oxidation and pACC in skeletal muscle following low intensity exercise (Dzamko et al., 2008). We demonstrated that carbohydrate ingestion during exercise resulted in lower skeletal muscle pACC (i.e. higher activity of ACC and therefore decreased fatty acid oxidation) compared to exercise in a fasted state or with niacin ingestion which was not apparently dependent on AMPK activation. This may the result of maintained aerobic rephosphorylation of ADP during exercise with carbohydrate ingestion resulting in reduced accumulation of AMP concentration and a reduction in glycogen utilisation, which itself may directly affect AMPK signalling (Gowans et al., 2013).

With carbohydrate ingestion, the concomitant changes in carbohydrate and fat availability limit the ability to know if carbohydrate or fatty acid availability drive exercise-induced signalling. We employed niacin ingestion to suppress fatty acid availability without stimulating insulin release or providing exogenous carbohydrate. Confirmation of this is shown with fasting circulating insulin concentrations and a lack of increased insulin signalling in muscle biopsies. We also observed an increase in muscle glycogen utilisation with niacin ingestion suggesting an increased reliance on glycogenolysis and an increase in pACC. It is assumed that glycogen concentrations regulate AMPK activation via a glycogen-binding domain on the $\beta 1$ subunit of AMPK potentially resulting in greater access of AMP to AMPK when glycogen concentrations are low (Steinberg et al., 2006). Although the effect of low glycogen availability on AMPK activity has been well researched with studies using the 'train-low' model in both acute and chronic settings (Impey et al., 2018), until now, no study has isolated carbohydrate and fatty acid availability because the reduction in carbohydrate availability is always accompanied with an increase in fatty acid availability.

We demonstrated similar pAMPK/AMPK ratios in all conditions most probably reflecting the effect of exercise, but greater pACC in both overnight fasted and niacin conditions, which both resulted in low carbohydrate availability. Niacin ingestion, however, also reduced fatty acid availability and whole-body fat oxidation compared to the overnight fasted state, suggesting that, although exercise is the main driver of AMPK activation, carbohydrate availability may independently regulate ACC under the conditions tested in the present study (Brownsey et al., 2006).

Carbohydrate ingestion before and during exercise results in increased carbohydrate availability, and low systemic fatty acid availability via insulin inhibition of lipolysis (Edinburgh et al., 2021). Following the onset of exercise, we observed a lower glycaemia, probably reflecting increased glucose flux into the muscle because of the concomitant actions of insulinaemia and muscle contraction (Goodyear et al., 1996). Therefore, although circulating glucose concentrations were low at this timepoint, the flux and availability of glucose to the muscle was still probably higher than with niacin ingestion and fasted exercise. This was reflected by the higher carbohydrate oxidation in the carbohydrate trial. In the CARB group, we also observed increased plasma insulinaemia and decreased NEFA concentrations (and whole-body fat oxidation). Furthermore, at the tissue level, we detected increased insulin signalling in skeletal muscle with an increase in pAkt with carbohydrate ingestion compared to fasted-state exercise. Despite greater pAkt with carbohydrate ingestion, we did not observe differences in INSR, which is probably because of the similar insulin concentrations between conditions at 120 min. We saw no difference in pAS160 or GLUT4 content between the conditions with little change pre- to post-exercise in both. Previous research has reported phosphorylation of AS160 on multiple sites in response to exercise (Vendelbo et al., 2014); therefore, a greater response may have been demonstrated probing for different or multiple phosphorylation sites. Moreover, GLUT4 content was similar before and after exercise and, although an acute bout of exercise with low glycogen concentration can increase GLUT4 translocation to the cell membrane (Derave et al., 1999), an acute bout of exercise will probably not increase total GLUT4 protein content.

It has been suggested that the activation of PDK4 and subsequent deactivation of PDH is reflective of carbohydrate utilisation in oxidative pathways (Zhang et al., 2014). PDH is part of the pyruvate dehydrogenase complex that plays a central role in glycolysis by catalysing pyruvate to acetyl-CoA for entry into the TCA cycle (Zhang et al., 2014). PDK4 is a regulatory enzyme of PDH and inactivates PDH via phosphorylation. Thus, inactivation of PDH via up-regulation of PDK4 can switch

substrate use from carbohydrate to fat oxidation. We saw no difference in total PDK4 content or pPDH[Ser293] between the conditions despite greater carbohydrate oxidation with carbohydrate and niacin ingestion. Previous research has demonstrated that carbohydrate ingestion prior to exercise increases exercise-induced activation of PDH (Tsintzas et al., 2000). Moreover, a similar niacin dose has been shown to increase PDH activation during exercise along with an increase in carbohydrate oxidation (Stellingwerff et al., 2003). However, although not significantly different from the carbohydrate condition, our data show a reduction in PDK4 content from baseline with niacin ingestion, which would favour an increase in PDH activation. Conversely, previous research demonstrates exercise with high fatty acid availability and/or low carbohydrate availability has been shown to increase PDK4 activation and transcription, and attenuate PDH activity (Hearris et al., 2018; Kiilerich et al., 2010; Pilegaard et al., 2002). Combined, these studies demonstrate a key role of insulin and glucose on PDH activation and substrate utilisation.

The lack of difference in pAMPK content between conditions in the present study suggests that exercise rather than carbohydrate availability may be a greater driver of AMPK activation. Similarly, the greatest differences in the mRNA transcriptome analysis were seen pre- *vs*. post-exercise, rather than between conditions. Combined, these suggest most of the changes in exercise-induced signalling at the transcriptomic level within muscle, may originate from exercise *per se* and not necessarily changes in carbohydrate or fatty acid availability and metabolism. However, the intensity of exercise in the present study may not have been high enough to fully induce pAMPK and, as such, cause differences in pAMPK between conditions. Previous research demonstrated that 1 h of cycling at 40% $\dot{V}_{O_2}$ peak similarly failed to increase pAMPK but did result in increases in pACC (Chen et al., 2003). In addition, although we did not see difference in pAMPK content, it is possible that differences in AMPK activity may be apparent; therefore, further research should attempt to explore AMPK activity under these conditions and explore other downstream targets of AMPK such as TBC1D1.

Our data suggest that exercise-induced cell signalling may be dependent on the carbohydrate availability during exercise. Whether these differences in cell signalling result in differences in exercise performance or health outcomes was not explored. Many previous studies that have shown changes in AMPK signalling with low carbohydrate availability compared to high carbohydrate availability have found no evidence of differences in improvement of $\dot{V}_{O_{2max}}$ (Gejl et al., 2017; Morton et al., 2009; Nybo et al., 2009; van Proeyen et al., 2011) or time trial performance (Hulston et al., 2010; Nybo et al., 2009; Yeo et al., 2008) between groups, suggesting that any differences in cell signalling may not always cause subsequent changes in performance. Moreover, data from our RNA transcriptome analysis suggest that most changes in exercise-induced cell signalling are seen pre- to post-exercise rather than between conditions, suggesting that the exercise itself probably drives potential improvements in performance or health outcomes rather than the carbohydrate or fatty acid availability during the exercise.

## Limitations

There may also be other effects of niacin ingestion in skeletal muscle that are currently not identified but may influence gene expression and cell signalling. Research demonstrates niacin acutely reduces adipose tissue lipolysis and free fatty acid availability (Christie et al., 1996) but 4 weeks of niacin supplementation can also influence mRNA expression involved in fatty acid utilisation in skeletal muscle of rodents (Ringsei et al., 2013). Therefore, we cannot exclude the possibility that niacin may affect skeletal muscle metabolism and signalling independent of the suppression of fatty acid availability. Similarly, the calories obtained from the carbohydrate drink and the resultant difference in net energy balance in the carbohydrate trial compared to the niacin and fasted exercise trial may have also altered gene expression or cell signalling independent of exercise. A further limitation of the present study is the lack of female participants, potentially reducing the generalisability of the findings, and, similarly, the findings may not be generalisable across different exercise modes or intensities. Because the present study was an exploratory analysis of a larger study, the sample size is small and the study could be underpowered to detect changes for some outcomes; therefore, caution is warranted when interpreting the findings. Finally, the number of biopsies was kept to the minimum required to answer the research question with the cross-over design; nevertheless, additional biopsies (e.g. pre-exercise) could have added further insight into substrate utilisation.

## Conclusions

In summary, our data demonstrate that carbohydrate ingestion suppresses exercise-induced phosphorylation of ACC compared to niacin ingestion, despite similarly low circulating fatty acid concentrations and whole-body fat oxidation. Although an acute bout of exercise is the main driver of AMPK activation, our data suggest carbohydrate availability may regulate ACC phosphorylation independent of fatty acid availability and independently from AMPK activation. These data

suggest that substrate availability does not affect AMPK responses to a single bout of exercise but does affect the signalling of downstream proteins, such as ACC.

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

## Additional information

### Data availability statement

Data for this study are openly available at reasearchdata.bath.ac.uk and https://www.ncbi.nlm.nih.gov/geo.

### Competing interests

JTG has received research funding from BBSRC, MRC, British Heart Foundation, Clasado Biosciences, Lucozade Ribena Suntory, ARLA Foods Ingredients, Cosun Nutrition Center and the Fruit Juice Science Centre; is a scientific advisory board member to ZOE; and has completed paid consultancy for 6d Sports Nutrition, The Dairy Council, PepsiCo, Violicom Medical, Tour Racing Ltd and SVGC. For a full list of disclosures see https://sites.google.com/view/declarationsgonzalez/home. JAB is an investigator on research grants funded by BBSRC, MRC, NIHR, British Heart Foundation, Rare Disease Foundation, EU Hydration Institute, GlaxoSmithKline, Nestlé, Lucozade Ribena Suntory, ARLA foods, Cosun Nutrition Center, American Academy of Sleep

Medicine Foundation, Salus Optima (L3M Technologies Ltd) and the Restricted Growth Association; has completed paid consultancy for PepsiCo, Kellogg's, SVGC and Salus Optima (L3M Technologies Ltd); is an Advisor to StudySetGo Ltd; is Company Director of Metabolic Solutions Ltd; receives an annual honorarium as a member of the academic advisory board for the International Olympic Committee Diploma in Sports Nutrition; and receives an annual stipend as Editor-in Chief of International Journal of Sport Nutrition & Exercise Metabolism.

## Author contributions

L.B. and J.T.G. conceived and designed the research. L.B., A.M.C., B.S., K.H. and A.C. collected the data. L.B., A.M.C., B.S., J.B., A.C. and J.M. analysed the samples. L.B. and T.V.dS analysed the data, interpreted the results and drafted the figures. LB drafted the initial manuscript. All authors reviewed and revised the manuscript. All authors approved the final version of the manuscript submitted for publication.

## Funding

This study was funded by The University of Bath.

## Keywords

cell signalling, carbohydrate metabolism, exercise metabolism, lipid metabolism

## Supporting information

Additional supporting information can be found online in the Supporting Information section at the end of the HTML view of the article. Supporting information files available:

**Peer Review History**

