## [Peer Review History · The Journal of Physiology]

Isolating the effects of carbohydrate and lipid availability on exercise-induced skeletal muscle signalling in males

Louise Bradshaw, Alfonso Moreno-Cabañas, Bruno Spellanzon, Katie Marie Hutchins, Adam J Collins, Jariya Buniam, Thibaux Van der Stede, Joanne Mallinson, Kostas Tsintzas, Wim Derave, Jean-Philippe Walhin, James Betts, Francoise Koumanov, and Javier T Gonzalez

DOI: 10.1113/JP289864

Corresponding author(s): Javier Gonzalez (J.T.Gonzalez@bath.ac.uk)

The following individual(s) involved in review of this submission have agreed to reveal their identity: Jorn Trommelen (Referee #1)

Review Timeline:

Submission Date:	04-Aug-2025
Editorial Decision:	22-Aug-2025
Revision Received:	15-Oct-2025
Accepted:	23-Oct-2025

Senior Editor: Karyn Hamilton

Reviewing Editor: Robert Musci

Transaction Report:

Dear Dr Gonzalez,

Re: JP-RP-2025-289864 "Isolating the effects of carbohydrate and lipid availability on exercise-induced skeletal muscle signalling in males" by Louise Bradshaw, Alfonso Moreno-Cabañas, Bruno Spellanzon, Katie Marie Hutchins, Adam J Collins, Jariya Buniam, Thibaux Van der Stede, Joanne Mallinson, Kostas Tsintzas, Wim Derave, Jean-Philippe Walhin, James Betts, and Javier T Gonzalez

Thank you for submitting your manuscript to The Journal of Physiology. It has been assessed by a Reviewing Editor and by 2 expert referees and we are pleased to tell you that it is potentially acceptable for publication following satisfactory major revision.

REVISION CHECKLIST:

We look forward to receiving your revised submission.

Yours sincerely,

Karyn Hamilton
Senior Editor
The Journal of Physiology

REQUIRED ITEMS

- You must start the Methods section with a paragraph headed Ethical approval (https://jp.msubmit.net/cgi-bin/main.plex?form_type=display_requirements#methods).

Research must comply with The Journal's policies regarding animal experiments (<https://physoc.onlinelibrary.wiley.com/hub/animal-experiments>) and adherence to these policies must be stated in the manuscript.

Authors should confirm in their Methods section that their experiments were carried out according to the guidelines laid down by their institution's animal welfare committee, including an ethics approval reference number. The Methods section must contain a statement about access to food, water and housing, details of the anaesthetic regime: anaesthetic used, dose and route of administration, and method of killing the experimental animals.

- You must start the Methods section with a paragraph headed Ethical Approval. If experiments were conducted on humans, confirmation that informed consent was obtained, preferably in writing, that the studies conformed to the standards set by the latest revision of the Declaration of Helsinki and that the procedures were approved by a properly constituted ethics committee, which should be named, must be included in the article file. If the research study was registered (clause 35 of the Declaration of Helsinki), the registration database should be indicated, otherwise the lack of registration should be noted as an exception (e.g. The study conformed to the standards set by the Declaration of Helsinki, except for registration in a database). For further information see: <https://physoc.onlinelibrary.wiley.com/hub/human-experiments>.

- Your manuscript must include a complete Additional Information section, including competing interests; funding; author contributions and acknowledgements.

- Please upload separate high-quality figure files via the submission form.

- You must upload original, uncropped western blot/gel images (including controls) if they are not included in the manuscript. This is to confirm that no inappropriate, unethical or misleading image manipulation has occurred. These should be uploaded as 'Supporting information for review process only'. Please label/highlight the original gels so that we can clearly see which sections/lanes have been used in the manuscript figures. For more information, see: <https://physoc.onlinelibrary.wiley.com/hub/journal-policies#imagmanip>.

- Papers must comply with the Statistics Policy: <https://jp.msubmit.net/cgi-bin/main.plex?>

form_type=display_requirements#statistics.

In summary:

- If n {less than or equal to} 30, all data points must be plotted in the figure in a way that reveals their range and distribution. A bar graph with data points overlaid, a box and whisker plot or a violin plot (preferably with data points included) are acceptable formats.
- If $n > 30$, then the entire raw dataset must be made available either as supporting information, or hosted on a not-for-profit repository, e.g. FigShare, with access details provided in the manuscript.
- ' n ' clearly defined (e.g. x cells from y slices in z animals) in the Methods. Authors should be mindful of pseudoreplication.
- All relevant ' n ' values must be clearly stated in the main text, figures and tables.
- The most appropriate summary statistic (e.g. mean or median and standard deviation) must be used. Standard Error of the Mean (SEM) alone is not permitted.
- Exact p values must be stated. Authors must not use 'greater than' or 'less than'. Exact p values must be stated to three significant figures even when 'no statistical significance' is claimed.

- Please include an Abstract Figure file, as well as the Figure Legend text within the main article file. The Abstract Figure is a piece of artwork designed to give readers an immediate understanding of the research and should summarise the main conclusions. If possible, the image should be easily 'readable' from left to right or top to bottom. It should show the physiological relevance of the manuscript so readers can assess the importance and content of its findings. Abstract Figures should not merely recapitulate other figures in the manuscript. Please try to keep the diagram as simple as possible and without superfluous information that may distract from the main conclusion(s). Abstract Figures must be provided by authors no later than the revised manuscript stage and should be uploaded as a separate file during online submission labelled as File Type 'Abstract Figure'. Please also ensure that you include the figure legend in the main article file. All Abstract Figures should be created using BioRender. Authors should use The Journal's premium BioRender account to export high-resolution images. Details on how to use and access the premium account are included as part of this email.

- Please ensure that all figures and tables have a title and legend, and that they have been cited within the main article text.

EDITOR COMMENTS

Reviewing Editor: Ethics Concerns:

Please indicate whether or not written informed consent was provided by subjects.

Comments to the Author:

Thank you for your submission to the Journal of Physiology. Both reviewers have raised excellent questions and comments, all of which ought to be addressed.

Please be particularly attentive over the shared concerns of study design, data reporting, and statistical analysis. These comments need to be comprehensively addressed.

In addition to the concerns raised by the reviewers, please refer to the Journal of Physiology Statistics Policy regarding data reporting. Data must be presented as mean {plus minus} SD unless SEM is justified. Additionally, when uploading figures/images, please ensure image quality is high as per Journal policy.

Senior Editor:

Comments for Authors to ensure the paper complies with the Statistics Policy:

Please visit The Journal's statistics policy to ensure compliance if you choose to submit a revised manuscript. For example, I noted that in one place, you express variability as \pm 95% CI in another place you use SEM, and in many other cases, it isn't

clear. The policy indicates that SD is the preferred/required approach. In most cases, you provide precise p-values, consistent with the policy (thank you). Please double check that for the figures as well. Additionally, the policy requires representation of individual data points in the figures wherever possible (I noted that you do this in some of your graphs).

Comments to the Author:

Thank you for submitting your manuscript for consideration by The Journal of Physiology. As part of the peer review process, we recruited two Referees with expertise in this field of study. Each has provided very detailed feedback including important questions that would need to be addressed before they believe that the manuscript will exert the impact that we aim for with The Journal of Physiology publications. At this point, we would like to invite you to consider whether or not each point can be addressed. If it can, we welcome submission of point by point responses and a manuscript revised according to the peer feedback for continued consideration by The Journal. If you choose to submit a revised manuscript, please ensure that it is in full compliance with The Journal's statistics policy. Please also indicate whether or not the study participants provided **written** informed consent. Thank you for your interest in The Journal of Physiology!

REFeree COMMENTS

Referee #1:

The authors have used a niacin intervention to isolate the effect of carbohydrate availability from lipid availability on muscle signaling following exercise. The study is novel and likely to be of interest to the readership. My primary comments concern 1) clarifying that the current study is part of a larger investigation, and 2) addressing the potential impact of the hypoglycemia observed in the CARB treatment. My comments are relatively minor and I am confident that the authors can address these points to further strengthen the manuscript. Please see my specific comments below.

Line 132. The trial registration appears to describe a larger study in which GLP-1 and energy intake following exercise are the primary outcomes. Those results do not appear to have been published yet, and the outcomes in the current manuscript do not appear to be registered. Could the authors clarify why the current data are being published separately rather than as part of the original research question outlined in the trial registration? This would improve transparency for readers and reviewers.

Line 132. (Follow up from previous comment): Please include a statement such as: "the current data are part of a larger investigation examining the impact of lipolysis on energy intake following exercise, as described in [registration number]". This will help readers locate related publications and reduce the risk that the current manuscript is treated as entirely independent in future publications in reviews and/or meta-analysis.

Line 278. A sample size justification is provided based on expected effect size. However, based on the trial registration, it appears that the current research question was not the original aim of the study. Was the main study's sample size truly determined for the outcomes presented in this manuscript, or were 8 participants from the larger trial analyzed here using this calculation? Please clarify. Given the absence of prospective registration for this research question, I recommend describing the study as an exploratory analysis from a larger project, and omitting the formal sample size justification. This would align with viewpoints expressed by one of the authors and others, that mechanistic physiological studies like the present work are inherently exploratory.

Line 287. (Follow up from previous comment): a statement is made about a priori comparisons. While the analysis seems appropriate, the a priori justification should be removed because it is not verifiable in the registration. The statement could be revised to: "to retain statistical power in this exploratory analysis, comparisons were limited...."

Line 192. Figure 1. Why were no samples collected at t=60 min (immediately before drink, supplement and the start of the exercise protocol)? This timepoint could serve as a pre-exercise baseline.

Figure 2C. The CARB treatment appears to induce severe reactive hypoglycemia. Please discuss these findings, including potential effects on muscle signaling, and whether similar outcomes would be expected if plasma glucose were comparable across conditions or elevated in the CARB treatment. You may also consider discussing glucose flux in this context.

Referee #2:

JP-RP-2025-289864 "Isolating the effects of carbohydrate and lipid availability on exercise-induced skeletal muscle signaling in males"

Overall: The premise of the work is low CHO or high NEFA may drive exercise adaptation when exercise is performed in the fasted state. Niacin was used to lower NEFA. In general findings show fat oxidation and NEFA levels are lower when CHO is consumed vs. fasting, although there was no difference with Niacin. Niacin did though increase reliance on muscle glycogen vs. CHO ingestion. There was also no effect on AMPK, although ACC-p was the results show no difference in NEFA between CHO and Niacin. In aggregate, the findings do showcase some differences between CHO intake vs. Niacin and supports ideas that CHO availability is a factor modulating adaptation to exercise. However, the statistical approach raises concern for type 1 error rates, thereby impacting interpretation of the findings.

Comments:

Title: signalling is spelled incorrectly

Abstract

NA

Introduction

NA

Methods and Results

Why were women not enrolled? This is a major weakness of the study and limits generalization. Appreciate the time frame of a within design, however, it would have been fair to test over 3 months and maintain phase for comparisons.

The age and BMI range is quite large. While a within design study, it would be of interest to know if these factors impacted responses to CHO or niacin vs. fasted exercise.

Was physical activity background considered (e.g. < 150 min/wk).

How was diet standardized and what was the macronutrient breakdown of such food? Were people fed to energy balance and this maintained across all 3 conditions?

Participants VO₂max and LT should be reported along with DXA information (e.g. body fat and lean mass). Moreover, submax, VO₂ data, energy expenditure, HR and RPE ought to be presented in a table (or supplement; unless it is being missed) so it can be interpreted. As is, statistics are shown for HR and RPE (Ln 352).

Also, age, BMI, and general demographics are not observed in the work. This should be reported in the manuscript.

Why does glucose drop at 75min in the CHO group?

In general with substrates, was fasting measures (0 min) compared across conditions to ensure no difference? Numerically it seems 0 min differs across groups, raising question as to whether results are over-estimated, and co-varying for fasted/0min values would provide more accurate assessments. Can the authors comment?

Given the weight discrepancy from inclusion criteria, how do substrate oxidation data differ if scaled to body weight? Also, was fasted fuel use obtained prior to drinks provided?

It is not clear why fuel use, protein content or glycogen level condition outcomes were measured by paired t-test. It would be more appropriate with this within subject design to assess by one-way ANOVA first. If a difference was identified between conditions, then a pairwise comparison, with correction (e.g. even FDR or B & H post-hoc), would be appropriate. As such, interpretation of the results is limited at this time and comments are truncated given the results may change based on the statistical approach.

Discussion

Ln 435-437, suggests AMPK is impacted by CHO availability. While it is appreciated that ACCp is a downstream regulated protein of AMPK, the results show no AMPK effects of the conditions. As a result, this conclusion or suggestion seems inaccurate. It would seem that the findings are potentially independent of AMPK. Throughout the discussion (e.g. Ln 452, etc.), modification of this point ought to be considered to reflect the findings.

The focus of this introduction is on adaptations. However, one cannot wonder though if exercise in the fasted state yields differential performance or health outcomes. While not the purpose of the work, it seems relevant to discuss despite potential mechanistic differences there may not be clear performance benefit or health gains based on mixed literature. Please insert such ideas to help communicate the application of these mechanistic findings.

The RNA seq data is somewhat difficult to discern with the number of subjects. But, with this in mind, and the general results, do the authors think that the act of feeding calories (CHO or fat or protein) in of itself is a factor altering gene expression/protein activation for adaptation independent of the nutrient? This is of course speculative in nature, but it is

interesting to think that the CHO group consumed kcal during exercise and reduced the level of energy deficit from exercise. The difference of this net energy balance then between Niacin or fast could be a factor also influencing the responses at gene or signaling levels of outcomes measured.

A limitations section should be included, namely small sample size, exercise intensity at LT for max fat oxidation thereby limiting generalization across exercise modes, and no women studied.

END OF COMMENTS

Responses to Reviewers Comments

We would like to thank the reviewers for taking the time to review the paper and for their insightful comments. Answers are shown in blue with the in text changes shown in red.

REQUIRED ITEMS

- You must start the Methods section with a paragraph headed **Ethical Approval**. If experiments were conducted on humans, confirmation that informed consent was obtained, preferably in writing, that the studies conformed to the standards set by the latest revision of the Declaration of Helsinki and that the procedures were approved by a properly constituted ethics committee, which should be named, must be included in the article file. If the research study was registered (clause 35 of the Declaration of Helsinki), the registration database should be indicated, otherwise the lack of registration should be noted as an exception (e.g. The study conformed to the standards set by the Declaration of Helsinki, except for registration in a database). For further information see: <https://physoc.onlinelibrary.wiley.com/hub/human-experiments>.

We have added in an ethical approval section.

- Your manuscript must include a complete **Additional Information section**, including competing interests; funding; author contributions and acknowledgements.

We have added in the subtitle for the additional information section.

- Please upload separate high-quality **figure files** via the submission form.

These files have been uploaded.

- You must upload original, uncropped western blot/gel images (including controls) if they are not included in the manuscript. This is to confirm that no inappropriate, unethical or misleading image manipulation has occurred. These should be uploaded as 'Supporting information for review process only'. Please label/highlight the original gels so that we can clearly see which sections/lanes have been used in the manuscript figures. For more information, see: <https://physoc.onlinelibrary.wiley.com/hub/journal-policies#imagmanip>.

These files have been uploaded.

- Papers must comply with the Statistics Policy: https://jp.msubmit.net/cgi-bin/main.plex?form_type=display_requirements#statistics.

In summary:

- If n {less than or equal to} 30, all data points must be plotted in the figure in a

way that reveals their range and distribution. A bar graph with data points overlaid, a box and whisker plot or a violin plot (preferably with data points included) are acceptable formats.

We have included all data points in figures that present summary statistics (e.g. bar graphs). However, where time series data are presented, we present the means and variance in line with recent papers published in J Phys (doi: 10.1113/JP289115). This is because all data points on a time series figure would be overly cluttered and difficult to interpret.

- If $n > 30$, then the entire raw dataset must be made available either as supporting information, or hosted on a not-for-profit repository, e.g. FigShare, with access details provided in the manuscript.

N/A

- 'n' clearly defined (e.g. x cells from y slices in z animals) in the Methods. Authors should be mindful of pseudoreplication.

n is clearly defined throughout

- All relevant 'n' values must be clearly stated in the main text, figures and tables.

As above. N is clearly defined throughout

- Please include an Abstract Figure file, as well as the Figure Legend text within the main article file. The Abstract Figure is a piece of artwork designed to give readers an immediate understanding of the research and should summarise the main conclusions. If possible, the image should be easily 'readable' from left to right or top to bottom. It should show the physiological relevance of the manuscript so readers can assess the importance and content of its findings. Abstract Figures should not merely recapitulate other figures in the manuscript. Please try to keep the diagram as simple as possible and without superfluous information that may distract from the main conclusion(s). Abstract Figures must be provided by authors no later than the revised manuscript stage and should be uploaded as a separate file during online submission labelled as File Type 'Abstract Figure'. Please also ensure that you include the figure legend in the main article file. All Abstract Figures should be created using BioRender. Authors should use The Journal's premium BioRender account to export high-resolution images. Details on how to use and access the premium account are included as part of this email.

This has been uploaded.

EDITOR COMMENTS

Reviewing Editor: Ethics Concerns:

Please indicate whether or not written informed consent was provided by subjects.

This has been added the ethical approval section at the start of the methods.

Comments to the Author:

Thank you for your submission to the Journal of Physiology. Both reviewers have raised excellent questions and comments, all of which ought to be addressed.

Please be particularly attentive over the shared concerns of study design, data reporting, and statistical analysis. These comments need to be comprehensively addressed.

In addition to the concerns raised by the reviewers, please refer to the Journal of Physiology Statistics Policy regarding data reporting. Data must be presented as mean {plus minus} SD unless SEM is justified. Additionally, when uploading figures/images, please ensure image quality is high as per Journal policy.

Senior Editor:

Comments for Authors to ensure the paper complies with the Statistics Policy:

Please visit The Journal's statistics policy to ensure compliance if you choose to submit a revised manuscript. For example, I noted that in one place, you express variability as +/- 95% CI in another place you use SEM, and in many other cases, it isn't clear. The policy indicates that SD is the preferred/required approach. In most cases, you provide precise p-values, consistent with the policy (thank you). Please double check that for the figures as well. Additionally, the policy requires representation of individual data points in the figures wherever possible (I noted that you do this in some of your graphs).

All data reporting has been changed to mean \$\pm\$ SD.

Comments to the Author:

Thank you for submitting your manuscript for consideration by The Journal of Physiology. As part of the peer review process, we recruited two Referees with expertise in this field of study. Each has provided very detailed feedback including important questions that would need to be addressed before they believe that the manuscript will exert the impact that we aim for with The Journal of Physiology publications. At this point, we would like to invite you to consider whether or not each point can be addressed. If it can, we welcome submission of point by point responses and a manuscript revised according to the peer feedback for continued consideration by The Journal. If you choose to submit a revised manuscript, please ensure that it is in full compliance with The Journal's statistics policy. Please also indicate whether or not the study participants provided written informed consent. Thank you for your interest in The Journal of Physiology!

REFEREE COMMENTS

Referee #1:

The authors have used a niacin intervention to isolate the effect of carbohydrate availability from lipid availability on muscle signaling following exercise. The study is novel and likely to be of interest to the readership. My primary comments concern 1) clarifying that the current study is part of a larger investigation, and 2) addressing the potential impact of the hypoglycemia observed in the CARB treatment. My comments are relatively minor and I am confident that the authors can address these points to further strengthen the manuscript. Please see my specific comments below.

Line 132. The trial registration appears to describe a larger study in which GLP-1 and energy intake following exercise are the primary outcomes. Those results do not appear to have been published yet, and the outcomes in the current manuscript do not appear to be registered. Could the authors clarify why the current data are being published separately rather than as part of the original research question outlined in the trial registration? This would improve transparency for readers and reviewers.

We agree with the reviewer and this has been amended as per the responses to the following comments. In brief, the biopsies were optional as part of the main study and many participants elected not to have the biopsies them therefore this is a sub-study of the larger study with different outcomes which is currently prepared for publication. We collectively feel that the findings are still worth reporting and will be of interest to the J Phys readers.

Line 132. (Follow up from previous comment): Please include a statement such as: "the current data are part of a larger investigation examining the impact of lipolysis on energy intake following exercise, as described in [registration number]". This will help readers locate related publications and reduce the risk that the current manuscript is treated as entirely independent in future publications in reviews and/or meta-analysis.

This has been added to the ethical approval paragraph at the start of the methods section (lines 128-129). This now reads "The current data are part of a larger investigation examining the impact of lipolysis on energy intake following exercise which is registered at clinicaltrials.gov (NCT05417659)."

Line 278. A sample size justification is provided based on expected effect size. However, based on the trial registration, it appears that the current research question was not the original aim of the study. Was the main study's sample size truly determined for the outcomes presented in this manuscript, or were 8 participants from the larger trial analyzed here using this calculation? Please clarify. Given the absence of prospective registration for this research question, I recommend describing the study as an exploratory analysis from a larger project, and omitting the formal sample size justification. This would

align with viewpoints expressed by one of the authors and others, that mechanistic physiological studies like the present work are inherently exploratory.

The biopsies were optional as part of the main study and 8 participants agreed to have the biopsies taken leading to a smaller sample size. This is also why the participants for this analysis are all male as no females opted to have muscle biopsies. This has now been added as a limitation at the end of the discussion (lines 661-663).

Line 287. (Follow up from previous comment): a statement is made about a priori comparisons. While the analysis seems appropriate, the a priori justification should be removed because it is not verifiable in the registration. The statement could be revised to: "to retain statistical power in this exploratory analysis, comparisons were limited...."

This has been changed in the statistical analysis section (lines 307-309). This now reads "To retain statistical power in this exploratory analysis, comparisons were limited to differences between two conditions".

Line 192. Figure 1. Why were no samples collected at t=60 min (immediately before drink, supplement and the start of the exercise protocol)? This timepoint could serve as a pre-exercise baseline.

Additional samples were not collected at these time points to limit participant burden according to comments at the ethical approval stage of the project. We have now added this as a limitation (lines 666-668).

"Finally, the number of biopsies were kept to a minimum required to answer the research question with the crossover design, nevertheless, additional biopsies (e.g. pre-exercise) could have added further insight into substrate utilisation."

Figure 2C. The CARB treatment appears to induce severe reactive hypoglycemia. Please discuss these findings, including potential effects on muscle signaling, and whether similar outcomes would be expected if plasma glucose were comparable across conditions or elevated in the CARB treatment. You may also consider discussing glucose flux in this context.

The lower glycaemia probably reflects increased flux into muscle from insulin and exercise stimulus for muscle glucose uptake. Therefore, whilst circulating glucose concentrations were low at this timepoint, the flux and availability of glucose for muscle was still likely higher than with niacin and fasted (represented by high CHO oxidation rates). This has been added to the discussion as below. The following has been added to the discussion (lines 574-579):

"Following the onset of exercise we observed a lower glycaemia, likely reflecting increased glucose flux into the muscle due to the concomitant plasma insulinemia and muscle contraction (Goodyear *et al.*, 1996). Therefore, whilst circulating glucose concentrations were low at this timepoint, the flux and availability of glucose to the muscle was still likely higher than with niacin ingestion and fasted exercise. This was

observed in the higher carbohydrate oxidation in the carbohydrate trial.”

Referee #2:

JP-RP-2025-289864 "Isolating the effects of carbohydrate and lipid availability on exercise-induced skeletal muscle signaling in males"

Overall: The premise of the work is low CHO or high NEFA may drive exercise adaptation when exercise is performed in the fasted state. Niacin was used to lower NEFA. In general findings show fat oxidation and NEFA levels are lower when CHO is consumed vs. fasting, although there was no difference with Niacin. Niacin did though increase reliance on muscle glycogen vs. CHO ingestion. There was also no effect on AMPK, although ACC-p was the results show no difference in NEFA between CHO and Niacin. In aggregate, the findings do showcase some differences between CHO intake vs. Niacin and supports ideas that CHO availability is a factor modulating adaptation to exercise. However, the statistical approach raises concern for type 1 error rates, thereby impacting interpretation of the findings.

Comments:

Title: signalling is spelled incorrectly

We have used the UK spelling of signalling as J Phys is a UK journal.

Abstract

NA

Introduction

NA

Methods and Results

Why were women not enrolled? This is a major weakness of the study and limits generalization. Appreciate the time frame of a within design, however, it would have been fair to test over 3 months and maintain phase for comparisons.

The biopsies were optional as part of the main study and although females were recruited for that study none of them opted to have muscle biopsies. We appreciate that this is a limitation of this study and we have added this to a limitations section at the end of the discussion.

The age and BMI range is quite large. While a within design study, it would be

of interest to know if these factors impacted responses to CHO or niacin vs. fasted exercise.

We have explored correlations between age and BMI versus the difference from CARB to FAST and NIACIN of substrate oxidation and fold change of ACC which are shown below. No correlations were significant with only 2 being moderate, therefore it is unlikely that age or BMI impacted the responses.

	BMI		Age	
	r	p	r	p
Total fat ox	0.26	0.32	0.25	0.35
Total CHO ox	-0.33	0.21	-0.18	0.49
Fold change in ACC phosphorylation status	-0.12	0.69	0.38	0.2

Was physical activity background considered (e.g. < 150 min/wk).

Background physical activity was not measured but a subjective measure of physical activity was used in calculating the intensity of each stage of the submaximal preliminary exercise test. This has been added to the participant characteristics table (Table 1).

How was diet standardized and what was the macronutrient breakdown of such food? Were people fed to energy balance and this maintained across all 3 conditions?

Participants were asked to record food intake for 24 hours prior to the preliminary testing and to replicate this prior to the main trial days. The participants ate their habitual diet. Whether the participants were in energy balance during this day was not determined but due to them consuming the same diet before every trial participants attended the laboratory in a similar nutritional state for every trial. Lines 150 -152 read:

“Participants recorded all dietary intake during the 24 hours prior to preliminary testing and were asked to repeat this intake prior to all subsequent visits to ensure participants were in a similar state of energy balance for all trials”

Participants VO₂max and LT should be reported along with DXA information (e.g. body fat and lean mass). Moreover, submax, VO₂ data, energy expenditure, HR and RPE ought to be presented in a table (or supplement; unless it is being missed) so it can be interpreted. As is, statistics are shown for HR and RPE (Ln 352).

VO₂ max was not measured. LT and DEXA information have been added to the participant characteristics table (Table 1). A table (Table 2) in the results section has been added to include VO₂, energy intake, HR and RPE data and statistics for energy intake and VO₂ have been added. A comment has been added in the relevant result section to this effect (lines 398-400).

“There were no significant differences in energy expenditure ($p = 0.91$) or $\dot{V}O_2$ ($p = 0.88$) between conditions.”

Also, age, BMI, and general demographics are not observed in the work. This should be reported in the manuscript.

A participant characteristics table (Table 1) has been added at line 145.

Why does glucose drop at 75min in the CHO group?

As per the response to reviewer 1, the lower glycaemia probably reflects increased flux into muscle from the combined effect of insulin and exercise stimulus for muscle glucose uptake. Therefore, whilst circulating glucose concentrations were low at this timepoint, the flux and availability of glucose for muscle was still likely higher than with niacin and fasted (represented by high CHO oxidation rates). This has been added to the discussion as below. The following has been added to the discussion (lines 574-579):

“Following onset of exercise we observed a lower glycaemia, likely reflecting increased glucose flux into the muscle due to the concomitant plasma insulinemia and muscle contraction (Goodyear *et al.*, 1996). Therefore, whilst circulating glucose concentrations were low at this timepoint, the flux and availability of glucose to the muscle was still likely higher than with niacin ingestion and fasted exercise. This was observed in the higher carbohydrate oxidation in the carbohydrate trial.”

In general with substrates, was fasting measures (0 min) compared across conditions to ensure no difference? Numerically it seems 0 min differs across groups, raising question as to whether results are over-estimated, and co-varying for fasted/0min values would provide more accurate assessments. Can the authors comment?

It is specifically recommended to not test for differences at baseline in randomised controlled trials (e.g. in CONSORT guideline; <https://doi.org/10.1016/j.jclinepi.2010.03.004> and <https://ijbnpa.biomedcentral.com/articles/10.1186/s12966-015-0162-z>). Since these tests are checking the probability that these differences occur at random, when it is known that they would be random at this timepoint. Indeed, the authoritative statistician Doug Altman has described this practice as “absurd” (<https://doi.org/10.2307/2987510>). As such we have not performed statistical tests for differences in baseline values as per the recommendation of renowned statisticians and CONSORT guidelines.

For the purposes of this review, however, we have explored the data with baseline values as covariates. The corrected and uncorrected means \pm 95 % CI are shown in the table below and this does not change the interpretation (n.b. “corrected” includes baseline as the covariate).

FAST	CARB	NIACIN
------	------	--------

	Uncorrected	Corrected	Uncorrected	Corrected	Uncorrected	Corrected
Total fat oxidation (g·60min⁻¹)	22.44 ± 7.22	22.64 ± 6.21	13.26 ± 6.35	13.15 ± 6.15	14.78 ± 6.78	14.69 ± 6.13
Total carb oxidation (g·60min⁻¹)	79.78 ± 17.84	78.9 ± 18.12	96.34 ± 19.8	96.57 ± 18.0	93.38 ± 22.39	94.02 ± 18.03

We have not done this for muscle glycogen or cell signalling data as the data are shown as change from baseline, therefore baseline data is already accounted for.

Given the weight discrepancy from inclusion criteria, how do substrate oxidation data differ if scaled to body weight? Also, was fasted fuel use obtained prior to drinks provided?

As this is a cross over design each participant is compared to themselves, therefore scaling the data for weight will make no difference to the inferences of comparisons between conditions. Furthermore, simple division of an outcome (e.g. fat oxidation in grams) by body mass (in kg) is subject to several limitations (<https://doi.org/10.1007/s40279-016-0655-1>). We therefore have not performed this adjustment. Fasted fuel use was obtained prior to the drinks or tablets being provided. There were no significant differences in fat or carbohydrate oxidation at this time point between the conditions.

It is not clear why fuel use, protein content or glycogen level condition outcomes were measured by paired t-test. It would be more appropriate with this within subject design to assess by one-way ANOVA first. If a difference was identified between conditions, then a pairwise comparison, with correction (e.g. even FDR or B & H post-hoc), would be appropriate. As such, interpretation of the results is limited at this time and comments are truncated given the results may change based on the statistical approach.

For the manuscript we stick with our decision to explore two comparisons as supported by the principle of closed loop testing (DOI: 10.1016/s0895-4356(00)00314-0) and by reviewer 1. For reassurance, however, we have assessed the data using a linear mixed model for these outcomes which show significant differences for these outcomes showing the inferences from the data remain the same (glycogen p = 0.04; pACC p = 0.02; pAkt p = 0.005).

Discussion

Ln 435-437, suggests AMPK is impacted by CHO availability. While it is appreciated that ACCp is a downstream regulated protein of AMPK, the results show no AMPK effects of the conditions. As a result, this conclusion or suggestion seems inaccurate. It would seem that the findings are potentially independent of AMPK. Throughout the discussion (e.g. Ln 452, etc.), modification of this point ought to be considered to reflect the findings.

We agree with the reviewer and have changed this point throughout the discussion and the abstract to reflect that the changes we see are independent of AMPK.

The focus of this introduction is on adaptations. However, one cannot wonder though if exercise in the fasted state yields differential performance or health outcomes. While not the purpose of the work, it seems relevant to discuss despite potential mechanistic differences there may not be clear performance benefit or health gains based on mixed literature. Please insert such ideas to help communicate the application of these mechanistic findings.

We agree with the reviewer and have added this to discussion on lines 609-622.

“Our data suggest exercise-induced cell signalling may be dependent on the carbohydrate availability during exercise. Whether these differences in cell signalling result in differences in exercise performance or health outcomes was not explored. Many previous studies that have shown changes in AMPK signalling with low carbohydrate availability compared to high carbohydrate availability have found no difference in improvement of VO₂max (Morton *et al.*, 2009; Nybo *et al.*, 2009; Van Proeyen *et al.*, 2011; Gejl *et al.*, 2017) or time trial performance (Yeo *et al.*, 2008; Nybo *et al.*, 2009; Hulston *et al.*, 2010) between groups suggesting that any differences in cell signalling may not cause subsequent changes in performance. Moreover, data from our RNA transcriptome analysis suggest that most changes in exercise-induced cell signalling are seen pre to post exercise rather than between conditions suggesting that the exercise itself likely drives potential improvements in performance or health outcomes rather than the carbohydrate or fatty acid availability during the exercise.”

The RNA seq data is somewhat difficult to discern with the number of subjects. But, with this in mind, and the general results, do the authors think that the act of feeding calories (CHO or fat or protein) in of itself is a factor altering gene expression/protein activation for adaptation independent of the nutrient? This is of course speculative in nature, but it is interesting to think that the CHO group consumed kcal during exercise and reduced the level of energy deficit from exercise. The difference of this net energy balance then between Niacin or fast could be a factor also influencing the responses at gene or signaling levels of outcomes measured.

We agree with the reviewer and have added this to the limitations section as per the answer below (lines 652-668).

A limitations section should be included, namely small sample size, exercise intensity at LT for max fat oxidation thereby limiting generalization across exercise modes, and no women studied.

We have added a limitations section to the end of the conclusion and have discussed these points (lines 652-668).

Dear Dr Gonzalez,

Re: JP-RP-2025-289864R1 "Isolating the effects of carbohydrate and lipid availability on exercise-induced skeletal muscle signalling in males" by Louise Bradshaw, Alfonso Moreno-Cabañas, Bruno Spellanzon, Katie Marie Hutchins, Adam J Collins, Jariya Buniam, Thibaux Van der Stede, Joanne Mallinson, Kostas Tsintzas, Wim Derave, Jean-Philippe Walhin, James Betts, Françoise Koumanov, and Javier T Gonzalez

We are pleased to tell you that your paper has been accepted for publication in The Journal of Physiology.

Required items:

1) Before we can continue with exporting your paper, we kindly request that you email us the article file with an Abstract Figure Legend included.

Yours sincerely,

Karyn Hamilton
Senior Editor
The Journal of Physiology

IMPORTANT POINTS TO NOTE FOLLOWING ACCEPTANCE OF YOUR PAPER:

- You can help your research get the attention it deserves! Check out Wiley's free Promotion Guide for best-practice recommendations for promoting your work at: www.wileyauthors.com/eoo/guide. You can learn more about Wiley Editing Services which offers professional video, design, and writing services to create shareable video abstracts, infographics, conference posters, lay summaries, and research news stories for your research at: www.wileyauthors.com/eoo/promotion.

- If you would like to receive our 'Research Roundup', a monthly newsletter highlighting the cutting-edge research published in The Physiological Society's family of journals (The Journal of Physiology, Experimental Physiology, Physiological Reports, The Journal of Nutritional Physiology and The Journal of Precision Medicine: Health and Disease), please click this link, fill in your name and email address and select 'Research Roundup': <https://www.physoc.org/journals-and-media/membernews>

EDITOR COMMENTS

Reviewing Editor:

Thank you for addressing all reviewer comments and making all the requested revisions. There is a minor comment from Reviewer 1 that authors should consider.

Senior Editor:

Thank you for submitting your revised manuscript for continued consideration by The Journal of Physiology. The Referees were both complimentary about the improvements resulting from the revisions. We are pleased to accept it for publication in The Journal of Physiology. Thank you for your interest in The Journal and Congratulations!

REFEREE COMMENTS

Referee #1:

The authors have responded well to my comments.

I agree that a $t=60$ min muscle biopsy would have increased the subject burden. However, I was wondering why no $t=60$ min plasma sample was collected? Plasma glucose shows a linear decline following carb ingestion, which may be an artefact of the sampling frequency. Is it likely that plasma glucose first increases and peaks 30-60 min following carbohydrate ingestion, and then shows a sharp decline following the start of exercise. If the authors agree, suggestions are to not connect the lines between the $t=0$ and $t=75$ min samples for this outcome and/or dedicate a sentence or two to speculate what happens during this period in the manuscript.

Referee #2:

Thank you for your responses. No further comments.